# WDR76 is a RAS binding protein that functions as a tumor suppressor via RAS degradation

Woo-Jeong Jeong[1,2], Jong-Chan Park[1,2], Woo-Shin Kim[1,2], Eun Ji Ro[1,2], Soung Hoo Jeon[1,2], Sang-Kyu Lee[1,2], Young Nyun Park[3], Do Sik Min[1,4] & Kang-Yell Choi[1,2]

Stability regulation of RAS that can affect its activity, in addition to the oncogenic mutations, occurs in human cancer. However, the mechanisms for stability regulation of RAS involved in their activity and its roles in tumorigenesis are poorly explored. Here, we identify WD40-repeat protein 76 (WDR76) as one of the HRAS binding proteins using proteomic analyses of hepatocellular carcinomas (HCC) tissue. WDR76 plays a role as an E3 linker protein and mediates the polyubiquitination-dependent degradation of RAS. WDR76-mediated RAS destabilization results in the inhibition of proliferation, transformation, and invasion of liver cancer cells. *WDR76*$^{-/-}$ mice are more susceptible to diethylnitrosamine-induced liver carcinogenesis. Liver-specific WDR76 induction destabilizes Ras and markedly reduces tumorigenesis in *HRas*$^{G12V}$ mouse livers. The clinical relevance of RAS regulation by WDR76 is indicated by the inverse correlation of their expressions in HCC tissues. Our study demonstrates that WDR76 functions as a tumor suppressor via RAS degradation.

[1] Translational Research Center for Protein Function Control, Yonsei University, Seoul, Korea. [2] Department of Biotechnology, College of Life Science and Biotechnology, Yonsei University, Seoul, Korea. [3] Department of Pathology, Yonsei University College of Medicine, Seoul, Korea. [4] Department of Molecular Biology, College of Natural Science, Pusan National University, Pusan, Korea. Correspondence and requests for materials should be addressed to K.-Y.C. (email: kychoi@yonsei.ac.kr)

RAS proteins (H, K, and NRAS) are small guanosine triphosphatases (GTPases) that play key roles in the regulation of pathophysiological processes including cell proliferation and transformation, and development[1,2]. The alternative binding states of GDP and GTP and membrane localization are well-known mechanisms controlling RAS proteins activity. The *RAS* mutations that fix RAS proteins as GTP binding forms occur in most human cancers[1,3–5]. In addition to the oncogenic mutations, the overexpression of RAS proteins that can also affect activity occurs in human cancers including colorectal cancer (CRC)[6–9] and a subset of breast cancers[10,11]. RAS elevation also occurs in HCCs; this elevation is associated with poor prognosis in patients[12–15]. Stabilization of RAS proteins constitutively activates downstream signaling pathways associated with tumorigenesis[6–8,16–20]. Particularly, in CRC, RAS stabilization via the Wnt/β-catenin pathway, especially by the *APC* mutations that are found in ~90% of human CRCs, plays important roles in the tumorigenesis[6–8]. In the resting state, RAS proteins are maintained at low levels due to proteasomal degradation by GSK3β-mediated phosphorylation and subsequent recruitment of the β-TrCP E3 linker protein[7,17]. In the case of aberrant Wnt/β-catenin signaling activation (e.g., caused by *APC* loss), RAS proteins and β-catenin are stabilized by inactivation of GSK3β, which results in enhancement of the colorectal tumorigenesis[7,21]. Especially, stabilization of mutant KRAS as well as β-catenin by *APC* loss is critical for the synergistic transformation of CRC[7,8]. Our investigation of RAS stability regulation by Wnt/β-catenin signaling revealed that some portion of RAS is degraded independently of the GSK3β-β-TrCP axis[7]. This result suggested the presence of an alternative mechanism for RAS stability regulation.

In this study, we use proteomic analysis to find proteins that interact with HRAS to identify other proteins involving degradation of RAS proteins independently of the GSK3β-β-TRCP system. We use purified GST-fused HRAS protein (GST-HRAS) as the bait for pull-down of HRAS binding partner proteins in tissue extracts from human HCC tumors, which express significantly higher levels of RAS compared with paired normal liver tissues. Potential HRAS binding proteins are separated using sodium dodecyl sulfate polyacrylamide gel electrophoresis (SDS-PAGE) and are subsequently identified by liquid chromatography tandem-mass spectrometry (LC/MS-MS) analyses. The validity of this experimental approach is confirmed by identification of proteins known to interact with RAS proteins. Next, we select proteins known to function in the ubiquitination-dependent degradation of proteins such as E3 ligases. We use knockdown of each of these candidate proteins and, identify WDR76, which is a CUL4-DDB1 ubiquitin E3 ligase interacting protein[22]. WDR76 was predicted to be a tumor suppressor candidate[23], and is a specific protein involved in degradation of RAS independently of the GSK3β-β-TRCP system.

Our in vitro studies reveal that RAS degradation mediated by WDR76 is directly related to the inhibition of proliferation, transformation, and invasion of liver cancer cells. We find that cytoplasmic WDR76 degrades RAS and mediates inhibition of cellular transformation. WDR76-mediated Ras degradation is verified using in vivo analyses comparing liver tissues from *HRas^{G12V}* and *HRas^{G12V}/WDR76* Transgenic (Tg) mice. *HRas^{G12V}*-driven liver carcinogenesis is significantly reduced in *HRas^{G12V}/WDR76* Tg mice, with a concomitant decrease in Ras protein levels and proliferation. The role of WDR76 as a tumor suppressor is also revealed by the high susceptibility to diethylnitrosamine (DEN)-induced inflammation, fibrosis, HCC progression, and lung metastasis in *WDR76^{−/−}* mice compared with those in wild-type (WT) mice. Moreover, the RAS staining intensities are much higher in tumor tissues compared with those

in paired non-tumor tissues and inversely correlate with the WDR76 staining intensities in human HCC tissues.

Taken together, our findings indicate that WDR76 is a suppressor of HCC tumorigenesis function via mediation of RAS degradation. Our studies present important evidence for a mechanism that controls RAS activity via regulation of its protein stability involving tumorigenesis and suggest that a mechanism regulating RAS stability via WDR76 may lead to development of strategies against human cancer.

## Results

**WDR76 is identified as one of the HRAS binding proteins.**
Overexpression of RAS proteins is frequently observed in patients with HCC, although most HCC tumors lack oncogenic *RAS* mutation[12–15]. Compared with the levels in paired normal tissues, the levels of RAS were significantly increased together with those of proliferating cell nuclear antigen (PCNA) in the tumor tissues from patients with HCC (Fig. 1a and Supplementary Fig. 1a). Gene set enrichment analyses showed enrichment for "MAPK pathway" and "signaling to RAS" gene sets in tumor tissues, compared with normal liver tissues (Fig. 1b). To identify HRAS binding partners involved in its degradation, we used tumor and non-tumor tissues from a patient with HCC with significantly increased RAS levels in tumor area. Affinity purification was performed against the human HCC or paired normal tissue extracts using purified recombinant GST-HRAS (with GST protein as a control), followed by separation of proteins by SDS-PAGE gel and then LC/MS-MS analysis was used to identify the proteins in parallel sliced 22 gel pieces for each lane of the gel (Fig. 1c). After subtraction of the common proteins that interacted with GST or beads, we identified 207 proteins (i.e., 108 from normal (N), 90 from tumor (T), and 9 from both N and T, tissue extracts). These proteins potentially interacted with HRAS and were covered various functional categories (Fig. 1d). The credibility of the experimental approach was confirmed by identification of 15 proteins known to interact with HRAS protein and 31 proteins were predicted to associate with HRAS, respectively (Supplementary Table 1). It is worth noting that previously known HRAS binding proteins including NF1[24], Fyn[25], JNK3[26], RASA2, and RASA4[27] were identified during our screen. Among the potential HRAS binding proteins, we confirmed HRAS binding characteristics of several candidate proteins that participate in signal transduction, including R-spondin3 (Rspo3)[28], Rab5[29], WD repeat domain 76 (WDR76)[22], and Interleukin 1 receptor like 1 (IL1RL1)[30] (Fig. 1e). To identify protein(s) involved in stability regulation of RAS, knockdown effects were examined for the several candidate HRAS binding proteins already known to be involved in ubiquitination-dependent protein stability regulation of proteins (e.g., WDR76[22], hect domain containing E3 ubiquitin protein ligase 1 (HectD1)[31], ubiquitination factor E4B (UBE4B)[32], ubiquitin specific peptidase 42 (USP42)[33], ubiquitin protein ligase E3B (UBE3B)[34], and ubiquitin specific peptidase 29 (USP29)[35]). The RAS levels were specifically upregulated by the small interfering RNA (siRNA) for WDR76, but not by HectD1, UBE4B, USP42, UBE3B, or USP29 (Fig. 1f). Both endogenous and exogenous RAS interacted with WDR76 regardless of its mutational status (i.e., oncogenic V12 or dominant-negative N17 mutations), as shown by immunoprecipitation in HEK293 cell extract (Fig. 2a and Supplementary Fig. 1b). Endogenous WDR76 was also pulled-down with all the three major RAS isotypes (Fig. 2b).

**WDR76 targets RAS for ubiquitination and degradation.**
Because WDR76 is an E3 ubiquitin ligase, we examined whether RAS proteins can be substrates for WDR76. Both endogenous and exogenous HRAS, KRAS, and NRAS protein levels were

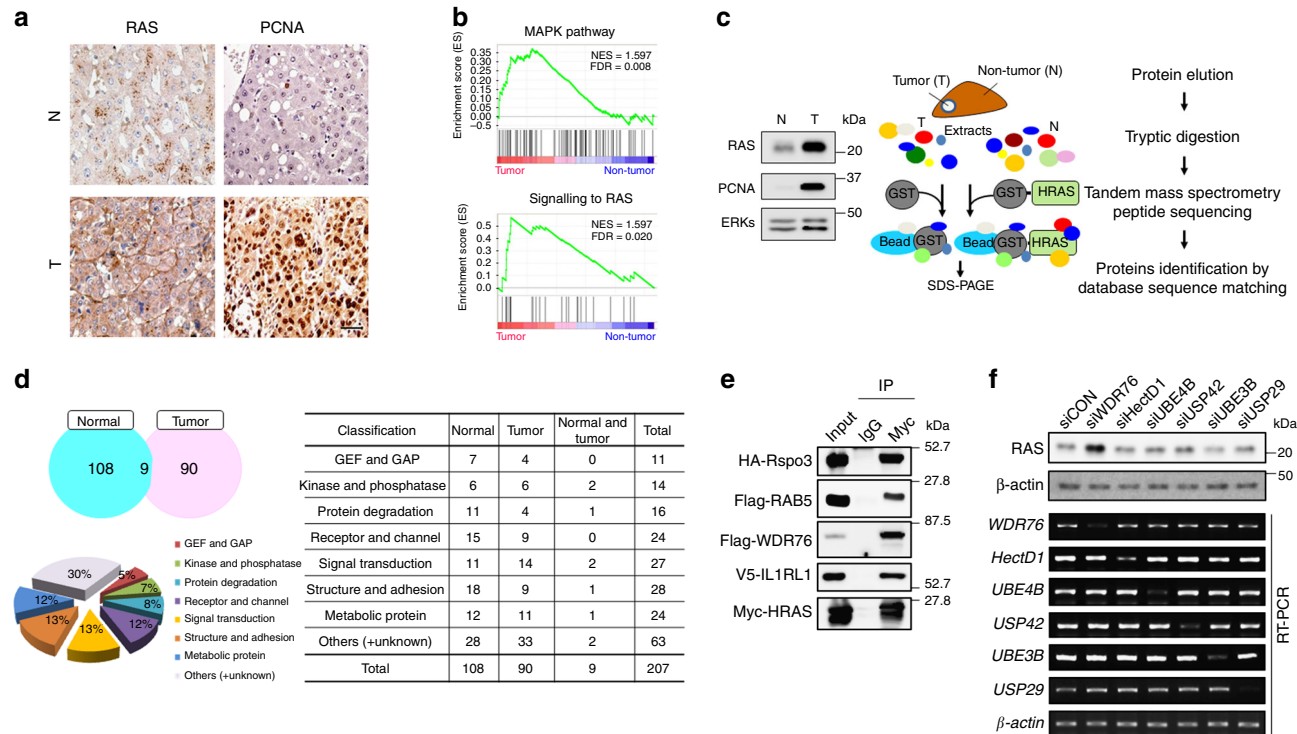

**Fig. 1** Identification of HRAS binding proteins. **a** Immunohistochemistry of RAS and PCNA in normal (N) and Tumor (T) tissues from patients with HCC. Scale bar, 100 μm. **b** GSEA profiles of "MAPK pathway" and "signaling to RAS" signatures between HCC tumors versus non-tumors. NES, normalized enrichment score; FDR, false discovery rate adjusted *p*-value. **c** Scheme to identify the HRAS interacting proteins. Total tissue lysates from a patient with HCC, identified RAS increment in tumor (T) compared with paired normal (N) tissues using immunoblot (IB) assay (left panel), were subjected to affinity purification using purified GST-HRAS or GST. Each purified protein complex was resolved on an SDS-PAGE gel and the bands were retrieved and analyzed by LC/MS-MS. **d** Classification of the identified HRAS interactomes. HRAS interactors identified using LC/MS-MS were categorized according to function. The numbers indicate identified HRAS interacting proteins for each category. **e** HEK293 cells were transfected with HA-Rspo3, Flag-RAB5, Flag-WDR76, or V5-IL1RL1, together with Myc-HRAS expression vector. Whole cell lysates (WCLs) were immunoprecipitated with anti-Myc antibody, and IB assay was performed with indicated antibodies. **f** HEK293 cells were transfected with siRNAs for *control* (CON), *WDR76, HectD1, UBE4B, USP42, UBE3B,* or *USP29*. The mRNA levels of each gene were detected by real-time polymerase chain reaction (RT-PCR). The RAS and β-actin were detected by IB

significantly reduced without changing their mRNAs by over-expression of WDR76 (Fig. 2c, d and Supplementary Fig. 1c, d). In contrast, WDR76 knockdown markedly increased endogenous RAS protein levels in a dose-dependent manner (Supplementary Fig. 1e). WDR76 overexpression accelerated the degradation rates of RAS, as shown by measurement in the presence of the de novo protein synthesis inhibitor cycloheximide (CHX) (Fig. 2e). The proteasome inhibitor *N*-acetyl-leucyl-leucyl-norleucinal (ALLN) reversed the WDR76-associated RAS reduction (Supplementary Fig. 1f). This result indicated that the RAS degradation by WDR76 occurred through the proteasomal machinery. RAS was polyubiquitinated by overexpression of WDR76 in both non-denaturing and denaturing conditions (Fig. 2f and Supplementary Fig. 1g). The enrichment of polyubiquitinated RAS proteins by overexpression of WDR76 was confirmed using the ubiquitin-specific UbiQapture-Q affinity matrices (Supplementary Fig. 1h). The polyubiquitination of RAS was further enhanced as it was further degraded by co-expression of WDR76 and Cul4A, which is a component of the CUL4-DDB1-WDR76 ubiquitin ligase complex[22] (Fig. 2g). Furthermore, we also confirmed the complex formation between WDR76/CUL4A and RAS (Supplementary Fig. 1i). We next tested whether WDR76 degraded oncogenic RAS mutants that frequently occur in cancer cells[3]. The levels of oncogenic mutant HRAS proteins (G12V and Q61L) and of WT were significantly reduced by overexpression of WDR76 (Supplementary Fig. 1j). The decreases and increases in oncogenic RAS by overexpression and knockdown of WDR76, respectively,

were further confirmed using the three cancer cell lines harboring a *RAS* mutation (i.e., T24T bladder cancer cell line ($HRAS^{G12V}$), LoVo colon cancer cell line ($KRAS^{G13D}$), and HepG2 hepatocel-lular carcinoma cell line ($NRAS^{Q61L}$)) (Fig. 2h, i).

**RAS degradation by WDR76 suppresses cellular transformation.** The effects of overexpression and knockdown of WDR76 on decrement and increment of RAS protein levels were verified, respectively, in Huh7, SK-Hep1, PLC/PRF/5, Hep3B, and HepG2 cells (Fig. 3a). WDR76 contains a carboxyl-terminal WD40 domain and was predicted to possess a nuclear localization signal (NLS; residues 199–208) (Fig. 3b). Because RAS proteins are localized mainly in the cytoplasm and plasma membrane, we hypothesized that RAS protein destabilization via WDR76 might occur in the cytoplasm. We evaluated the role of cytoplasmic WDR76 on RAS destabilization by generating an NLS deletion mutant (WDR76ΔNLS) (Fig. 3b). Both degradation and ubiqui-tination of RAS were significantly enhanced by the cytoplasmic localization of WDR76; compared with full-lengh (FL) WDR76, a greater amount of WDR76ΔNLS was co-immunoprecipitated by RAS (Supplementary Fig. 2a-c). To evaluate the role of RAS destabilization for regulation of liver cancer cell proliferation and transformation, we generated SK-HEP1 stable cell lines that overexpressed GFP-WDR76FL or GFP-WDR76ΔNLS (Sup-plementary Fig. 2d). Consistently, the level of RAS protein was more significantly reduced in SK-Hep1 cells overexpressing GFP-WDR76ΔNLS (Fig. 3c). Proliferation and transformation of SK-

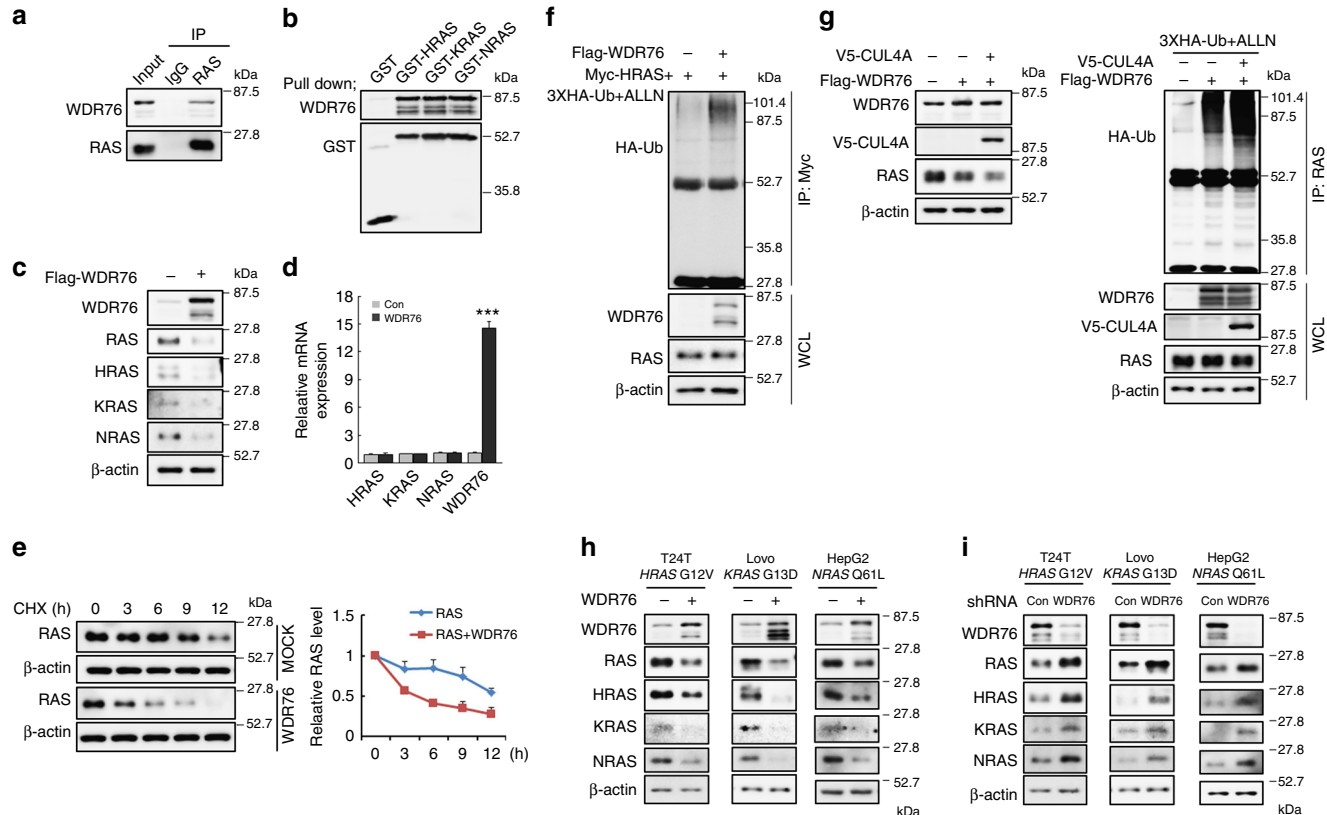

**Fig. 2** WDR76 interacts with RAS and mediates their ubiquitination-dependent proteasomal degradation. **a** Interaction of endogenous RAS with WDR76 determined by immunoprecipitation in HEK293 cells. **b** HEK293 WCLs were pulled-down with recombinant GST-H, -K, or -NRAS protein. **c–e** HEK293 cells were transfected with *Flag-WDR76* expression vector. WCLs were subjected to IB to detect indicated proteins (**c**). The mRNA levels of *H, K, NRAS*, and *WDR76* were measured by quantitative RT-PCR; geometric mean of *GAPDH*, *β-actin*, and *HPRT* was used as an internal control. Data are presented as the mean ± SD ($n = 3$ biological replicates). Statistical significances were assessed using two-sided Student's *t* test, ***$p < 0.001$ (**d**). The levels of RAS at indicated time points after CHX treatment were determined by IB, and were quantified with β-actin as a loading control. Results plotted on the the right are the amounts of RAS at each time point relative to the level at time 0. Data are presented as the mean ± SD ($n = 3$ biological replicates) (**e**). **f** HEK293 cells were transfected with indicated vectors and then treated with ALLN. WCLs were then immunoprecipitated with an anti-Myc antibody. **g** HEK293 cells were transfected with Flag-WDR76 or V5-CUL4A together with 3XHA-Ub construct then treated with ALLN (left panel). WCLs were immunoprecipitated with anti-RAS antibody. **h**, **i** T24T, Lovo, or HepG2 cells were transfected with WDR76 (**h**) or shWDR76 (**i**)

Hep1 cells were reduced by overexpression of GFP-WDR76FL, and these inhibitory effects were further enhanced in cells that expressed GFP-WDR76ΔNLS (Supplementary Fig. 2e, f). Overall, our results indicated that cytoplasmic WDR76 plays role in RAS destabilization and in suppression of liver cancer cells transformation. The role of WDR76-mediated RAS destabilization in suppression of proliferation was confirmed by generation of a WDR76-deficient SK-Hep1 (SK-KO) cell line using the clustered regularly interspaced short palindromic repeats (CRISPR)-Cas9 system, and subsequent characterization using proliferation assays (Supplementary Fig. 3a–d). The functional relevance of WDR76 and the greater efficiency of cytoplasmic WDR76 for the polyubiquitination-dependent degradation of RAS were further confirmed using reconstitution assays that revealed the rescue effects of the degradation and polyubiquitinylation of RAS, and the WDR76-RAS interaction in the SK-KO cell line (Fig. 3d–f). Consistent with the destabilization of RAS by WDR76, the anti-proliferation and anti-transforming effects of WDR76 were confirmed by comparing SK-WT and SK-KO cells and measuring the reconstitution effects of GFP-WDR76FL or GFP-WDR76ΔNLS on cell growth and foci formation (Fig. 3g, h). We next assessed whether WDR76 KO-induced cell transformation occurred via RAS proteins upregulation. Simultaneous knockdown of RAS strongly suppressed WDR76 KO-induced

transformation of the SK-KO cells (Supplementary Fig. 3e), confirming that the transformation induced by WDR76 KO was dependent on RAS. We tested whether WDR76 KO-induced cell transformation occurred via activation of the MAPK pathway by monitoring the effects of AS703026, a specific MEK inhibitor[36]. WDR76 KO-induced transformation was suppressed by AS703026 treatment (Supplementary Fig. 3f), indicating that RAS stabilization by WDR76 KO promotes cell transformation via the MAPK pathway.

Taken together, our results indicated that WDR76, especially WDR76 located in the cytoplasm, suppressed transformation of cells via degradation of RAS.

**RAS degradation by WDR76 suppresses cell invasive properties.** Transforming growth factor β (TGFβ) cooperates with HRAS to promote epithelial-mesenchymal transition (EMT) process that contributes to tumor invasion and metastasis[37–39]. Therefore, we examined the role of WDR76 in the TGFβ-induced EMT associated with RAS. Overexpression of WDR76 in SK-Hep1 cells reduced RAS levels and inhibited TGFβ-induced EMT phenotypes such as the increment of the mesenchymal marker N-cadherin (Supplementary Fig. 3g). WDR76 knockdown or HRAS overexpression further enhanced TGFβ-induced N-cadherin

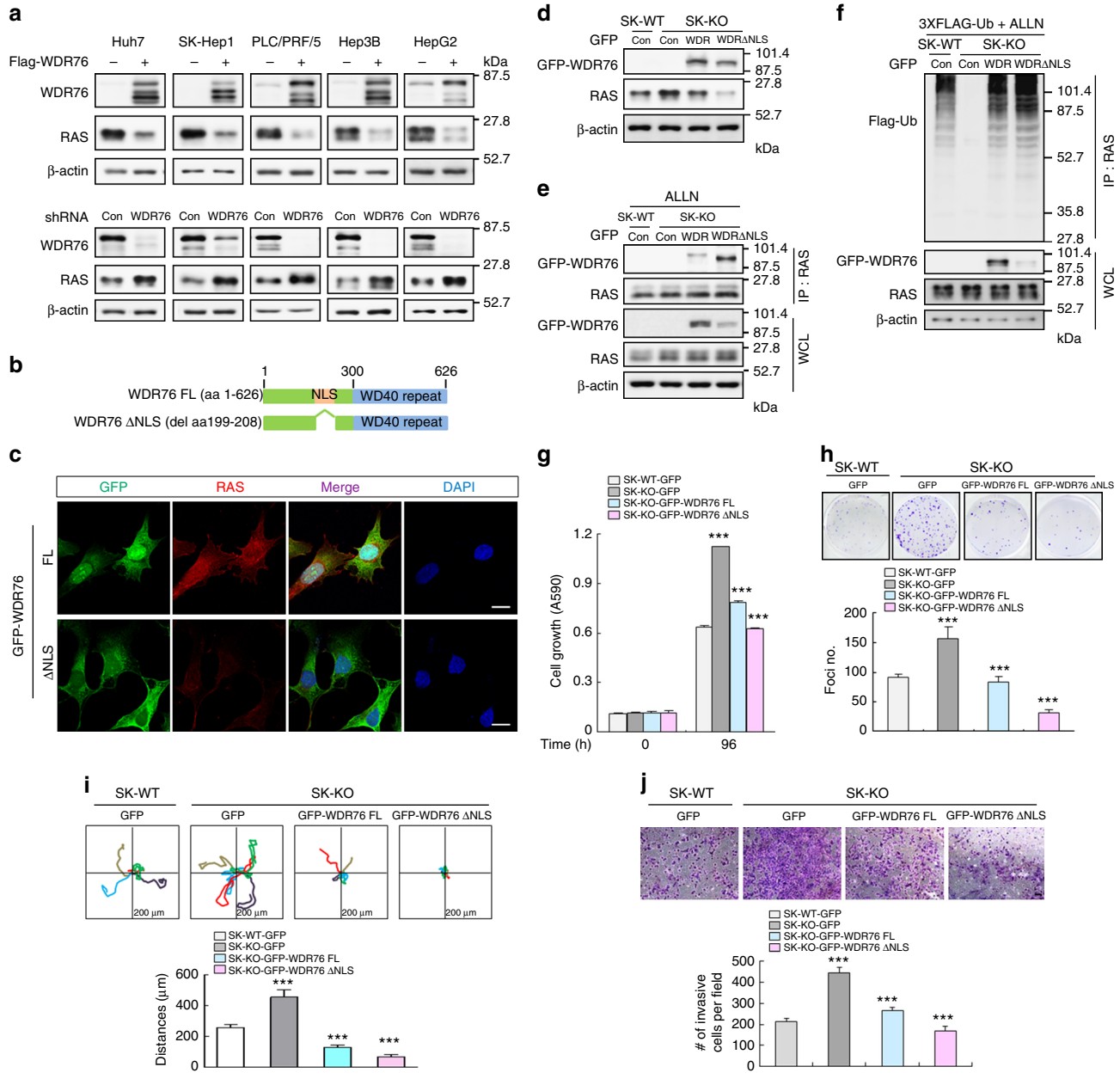

**Fig. 3** WDR76 suppresses proliferation, transformation, motility and invasive properties of liver cancer cells through RAS destabilization. **a** IBs of RAS in *Flag-WDR76* or *shWDR76*-transfected Huh7, SK-Hep1, PLC/PRF/5, Hep3B, and HepG2 cells. **b** Schematic representation of WDR76 and its NLS deleted forms. **c** SK-Hep1 cells stably expressing GFP-WDR76FL or GFP-WDR76ΔNLS were analyzed. Immunocytochemical analyses data showing localizations of RAS and WDR76 were obtained by using the anti-RAS (red) or anti-GFP (green) antibody. The nuclei were stained with DAPI (blue). Scale bars, 20 μm. **d–f** SK-WT cells stably expressing GFP, or SK-KO cells stably expressing GFP, GFP-WDR76FL, or GFP-WDR76ΔNLS were analyzed (**d**), or were transfected with 3XFlag-Ub (**f**) and then treated with ALLN (**e**, **f**). WCLs were immunoprecipitated with antibody recognizing RAS (**e**, **f**). **g**, **h** Cells were cultured and MTT assay (**g**), and foci formation assays (**h**) were performed. Data are presented as the mean ± SD ($n = 3$ biological replicates). Two-sided Student's $t$ test, ***$p < 0.001$. **i** Single-cell migratory behavior was monitored using real-time imaging microscopy at least five independent times. Data are presented as the mean ± SD. **j** Invaded cells through the Matrigel were stained with crystal violet. Representative images were captured, and the total numbers of invaded cells were counted (bottom). Data are presented as the mean ± SD ($n = 3$ biological replicates). Two-sided Student's $t$ test, ***$p < 0.001$. Scale bar, 200 μm

expression (Supplementary Fig. 3h, i). These EMT phenomena were increased, as shown by upregulation of N-cadherin via RAS stabilization in WDR76-deficient SK-KO-GFP cells. Reduction of RAS levels in SK-KO-GFP-WDR76FL and SK-KO-GFP-WDR76ΔNLS cells resulted in reversal of the EMT phenotype, which confirmed that the WDR76-associated modulation of the EMT phenotype was dependent on RAS (Supplementary Fig. 3j).

Accordingly, WDR76 deficiency increased the migration and invasiveness of SK-Hep1 cells; these responses were inhibited by reconstitution of WDR76 FL and more significantly, by that of WDR76ΔNLS (Fig. 3i, j).

**WDR76 loss promotes DEN-induced hepatocarcinogenesis.** *WDR76* knockout mice were generated to examine the in vivo

roles of WDR76 on Ras protein stability regulation (Supplementary Fig. 4a, b). *WDR76* knockout was also confirmed using in situ hybridization of mouse liver tissue and immunoblot analysis of various tissues (e.g., liver, colon, lung, spleen, stomach, and kidney) (Supplementary Fig. 4c, d). The *WDR76*[−/−] mice were viable, fertile, and had no obvious developmental abnormalities. We isolated mouse embryonic fibroblast cells (MEFs) from *WDR76*[+/+] and *WDR76*[−/−] mice, and found that endogenous HRas, KRas, and NRas protein levels were increased in *WDR76*[−/−] MEFs compared with *WDR76*[+/+] MEFs; the mRNA levels remained unchanged (Supplementary Fig. 4e). Loss of *WDR76* reduced the Ras degradation rates in *WDR76*[−/−] MEFs (Supplementary Fig. 4f). The effect of *WDR76* deficiency was rescued by re-introduction of *WDR76*, and Ras degradation via polyubiqutinylation-dependent proteasomal degradation was further confirmed (Supplementary Fig. 4g, h). In *WDR76*[+/+], but not *WDR76*[−/−] MEFs, the Ras levels were reduced with the increment of its polyubiquitination by CUL4A overexpression (Supplementary Fig. 4i, j). Concomitant with the increased Ras abundance, we found an increment of hepatocyte proliferation in liver tissues from 2-week-old *WDR76*[−/−] mouse (Supplementary Fig. 5a-c). Robust tumorigenesis was not found in the livers of 1-year-old *WDR76*[−/−] mice. However, severe fibrosis and collagen deposition were observed as evaluated histologically by Sirius red staining together with Ras levels increment. The positive staining for α-smooth muscle actin (αSMA), which was used as a marker for activated fibrogenic cells were also concomitantly increased (Supplementary Fig. 5d-g).

To determine any protective role of WDR76 during HCC pathogenesis, effects of *WDR76* knockout were investigated for HCC tumorigenesis induced by DEN (i.e., a representative hepatic carcinogen), which closely resembles histologic and genetic features of human HCC[40]. First, we assessed the liver injury induced by acute DEN treatment. Compared with WT mice, *WDR76*[−/−] mice were more sensitively responded to liver injury induced by acute DEN treatment as shown by higher serum alanine aminotransferase (ALT) levels, indicative of liver injury, and transcriptional induction of proinflammatory genes, such as tumor necrosis factor-α (*TNF-α*), and interleukin (*IL*)-1β (Supplementary Fig. 6a-c). DEN-induced compensatory proliferation was more pronounced in the *WDR76*[−/−] mice compared with the *WDR76*[+/+] mice as assessed by Ki67-positive cells (Supplementary Fig. 6d, e). To test whether *WDR76* deficiency-mediated Ras accumulation could enhance HCC development, 2-week-old *WDR76*[+/+] and *WDR76*[−/−] mice were injected with DEN[41,42] (Fig. 4a). WDR76 deficiency resulted in enhanced development of DEN-induced HCC phenotypes, including promoted tumor formation, and increment in liver weight and liver/body weight ratios (Fig. 4b, c). We consistently found greater numbers of Ki67-positive cells and higher Ras expression in *WDR76*[−/−] tumors (Fig. 4d-f). Notably, WDR76 deficiency resulted in increased sensitivity to hepatocarcinogenesis and an increased incidence of metastasis to the lung (Fig. 4g).

**HRas-driven hepatic carcinogenesis is suppressed by WDR76.** To further define the function of WDR76 in *HRas*[G12V]-induced hepatic carcinogenesis, we generated mutant mice carrying *HRas*[G12V] or *HRas*[G12V]/*WDR76* Tg mice, and examined Ras levels in the livers of 6-week-old mice. The effects of overexpression of WDR76 on decrement of RAS protein levels in vivo were verified (Supplementary Fig. 7a, b). At 52-weeks of age, significant decreases in liver weights and liver/body weight ratios which is an indicator of tumor burden were evident in *HRas*[G12V]/*WDR76* Tg mice compared with *HRas*[G12V] mice (Fig. 5a-c). Consistently, we observed decreased Ki67-positive cells, ERK

activity, and Ras levels without changing *HRas* mRNAs in *HRas*[G12V]/*WDR76* Tg mouse livers compared with livers from *HRas*[G12V] mice (Fig. 5d-f).

**Elevated RAS correlates with low WDR76 levels in human HCC.** To validate the possible tumor-suppressive role of WDR76 in human HCC, we analyzed a tissue microarray (TMA; LV1505; US Biomax) consisting of 46 cases of human hepatocellular carcinoma with paired adjacent non-tumor tissues. Immunohistochemical (IHC) analyses of these tissues revealed that RAS levels were higher in 41/46 (89.1%) HCC patient tumors compared with their adjacent non-tumor tissues, whereas WDR76 expression levels were lower in 34/46 (73.9%) of HCC tumors (Fig. 6a-c). Comparison of the protein expression ratios (tumor/non-tumor) of RAS and WDR76 from the regions of the tumor and paired non-tumor tissues further revealed a significant negative correlation between RAS and WDR76 (non-tumor; $r = -0.6671$, $p < 0.001$ and tumor; $r = -0.4573$, $p = 0.002$) (Fig. 6d). Thus, there is a significant association between RAS and WDR76 levels in human HCC.

**Discussion**
Although, activities of the RAS proteins have been mainly known to be regulated by the GTP and GDP loading switch and membrane localization, emerging evidences are suggesting that stability regulation of RAS also plays important roles in pathophysiology[6–8,16–18,21,43,44]. Recently, it has been reported that Nedd4-1 targets RAS proteins for their degradation and suppresses tumorigenesis, but activation of RAS evades the Nedd4-1-mediated degradation, leading to potential initiation of tumorigenesis[20]. The recent identification of a mechanism for RAS protein stability regulation via Wnt/β-catenin signaling[7] in CRC model provides an alternate approach for control of RAS activity via regulation of RAS protein stability[21]. However, a question remains whether RAS protein stability regulation via the Wnt/β-catenin signaling is a unique mechanism for RAS degradation or whether one or more other mechanisms contribute to the degradation of RAS protein.

Aberrant activation of the RAS-ERK pathway is involved in the progression of human HCC and complex mechanisms lead to activation of the RAS pathway[12]. The RAS pathway activation can be amplified by suppression of the negative regulators of RAS/MAPK pathway, including RASAL1, DAB2IP, NORE1A, RKIP, and SPRY2[13,45,46] or by proteasome-dependent degradations of the negative regulators of RAS/MAPK pathway such as RASSF1A and DUSP1[47] in human HCC.

Although oncogenic mutations of *RAS* are rare in patients with HCC, overexpression of RAS proteins is frequently observed and is associated with a poor prognosis[14]. To systematically identify HRAS binding proteins involved in its degradation, we used a proteomics approach with tumor and non-tumor tissues from a human patient with HCC. The RAS levels in tumor tissues were significantly increased compared with those in paired normal liver tissues. It is worth noting that previously known HRAS binding proteins including NF1[24], Fyn[25], JNK3[26], RASA2, and RASA4[27] were identified during our screen. We identified potential HRAS binding proteins that might be useful for better understanding of RAS proteins related with human cancer.

WDR76, a CUL4-DDB1 ubiquitin E3 ligase interacting protein, degraded RAS via polyubiquitination-dependent proteasomal degradation. The effects of overexpression and knockdown of WDR76 on decrement and increment of RAS protein levels, respectively, were verified in cancer cells including *RAS*-mutated cancer cells, indicating that WDR76 degrades RAS proteins regardless of their mutational status. The role of cytoplasmic

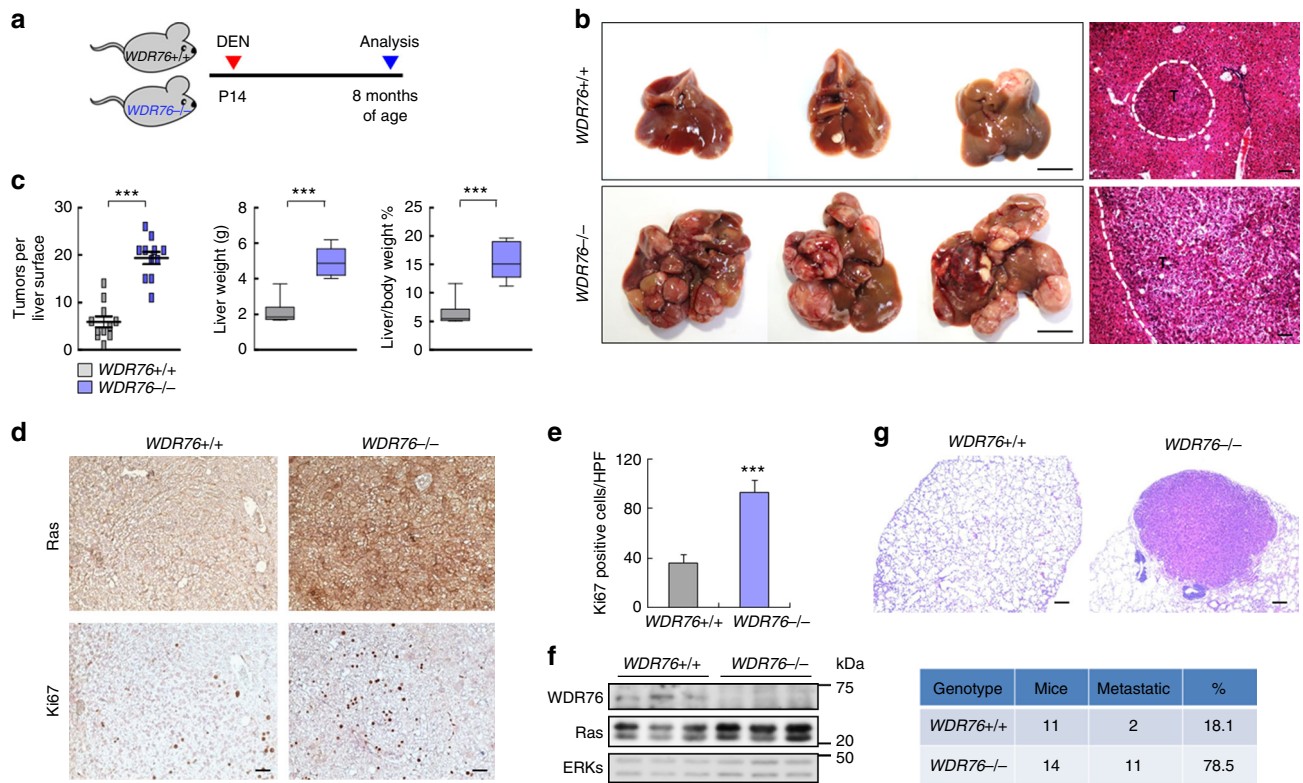

**Fig. 4** WDR76 deficiency results in Ras accumulation in liver and promotes DEN-induced HCC. **a** Schematic representation of DEN-induced HCC model. Two-week-old $WDR76^{+/+}$ and $WDR76^{-/-}$ mice were intraperitoneally injected with DEN and euthanized after 8 months. **b** Representative gross images (left; scale bars, 10 mm) and H&E staining (right; scale bars, 200 μm) of liver tissues of $WDR76^{+/+}$ and $WDR76^{-/-}$ mice described in (**a**). "T" denotes tumor, encircled by the dotted line. **c** Boxplots show the numbers of visible tumors on the liver surface, liver weights, and liver-to-body weight ratios ($n = 11$ WT and 11 KO). Data are presented as the mean ± SEM. Statistical significances were assessed using two-sided Student's $t$ test, ***$p < 0.001$. **d** IHC analyses for Ras and Ki67 in $WDR76^{+/+}$ and $WDR76^{-/-}$ liver tumor sections. Scale bars, 100 μm. **e** Quantification of the number of Ki67-positive cells per 10 high power field (HPF) in liver tissue sections. Data are presented as the mean ± SD. Statistical significances were assessed using two-sided Student's $t$ test, ***$p < 0.001$. **f** IBs of Ras and ERKs from liver tissues of $WDR76^{+/+}$ and $WDR76^{-/-}$ mice. **g** H&E images (top) and numbers (bottom) of lung metastases from $WDR76^{+/+}$ and $WDR76^{-/-}$ mice given DEN at P14. Scale bars, 500 μm

WDR76 in RAS degradation was confirmed by enhancement of its ubiquitination and degradation by WDR76ΔNLS. RAS stability regulation via WDR76 is directly related to various aspects of pathophysiology, including cell proliferation and transformation, and the invasive properties of liver cancer cells as shown by correlations of these aspects with the RAS protein level which are controlled by modulation of WDR76.

WDR76-mediated Ras degradation and its tumor-suppressive role were consistently verified by comparing liver tissues of $WDR76^{+/+}$ versus $WDR76^{-/-}$ mice and $HRas^{G12V}$ versus $HRas^{G12V}/WDR76$ Tg mice. We found an increment of hepatocyte proliferation with increment of Ras proteins levels in liver tissues from 2-week-old $WDR76^{-/-}$ mice. We also found severe hepatic fibrosis with Ras increment in livers from 1-year-old $WDR76^{-/-}$ mice, but not in livers from same age $WDR76^{+/+}$ mice.

Because HCC frequently develops in the setting of liver injury and inflammation, we administered DEN to induce HCC in mice. The acute liver injury caused by DEN exposure that accompanied the hepatocyte DNA damage and expression of inflammatory cytokines was significantly increased in $WDR76^{-/-}$ mice. Indeed, WDR76 deficiency enhanced the formation of malignant liver tumors and lung metastasis providing strong evidence that WDR76 is a potential tumor suppressor. Notably, we also found higher expression of Ras and greater numbers of Ki67-positive cells in $WDR76^{-/-}$ tumors. Thus, it is possible that Ras level

increment in the livers of the $WDR76^{-/-}$ mice contributed to the hepatocarcinogenesis. We further investigated these findings using the $HRas^{G12V}$ liver cancer model. Degradation of Ras in the $HRas^{G12V}$-induced HCC model by crossing $HRas^{G12V}$ mice with WDR76 Tg mice resulted in a significant reduction in tumor incidence. Taken together, these in vivo results here clearly present important evidence for a mechanism that controls RAS activity via regulation of its protein stability involving tumorigenesis. The pathological significance of the aberrancy in the RAS stability regulation via WDR76 was supported by the inverse correlation between expression levels of RAS and WDR76 in nontumor and tumor tissues of HCC patients. The RAS overexpression in HCC patients was also reported in previous studies[13,14].

The presented studies identify and characterize a RAS binding protein WDR76 which plays a role as an E3 linker protein and mediates polyubiquitination-dependent degradation of RAS involving suppression of HCC tumorigenesis and metastasis. Our findings provide pathophysiological implications that RAS activity can be controlled via regulation of protein stability which can affect suppression of the tumorigenesis and metastasis. Considering presence of the multiple mechanisms of the RAS stability regulation by different signaling pathways, regulation of RAS at the level of protein stability also could be a general strategy of cells for the modulation of complex extracellular stimuli.

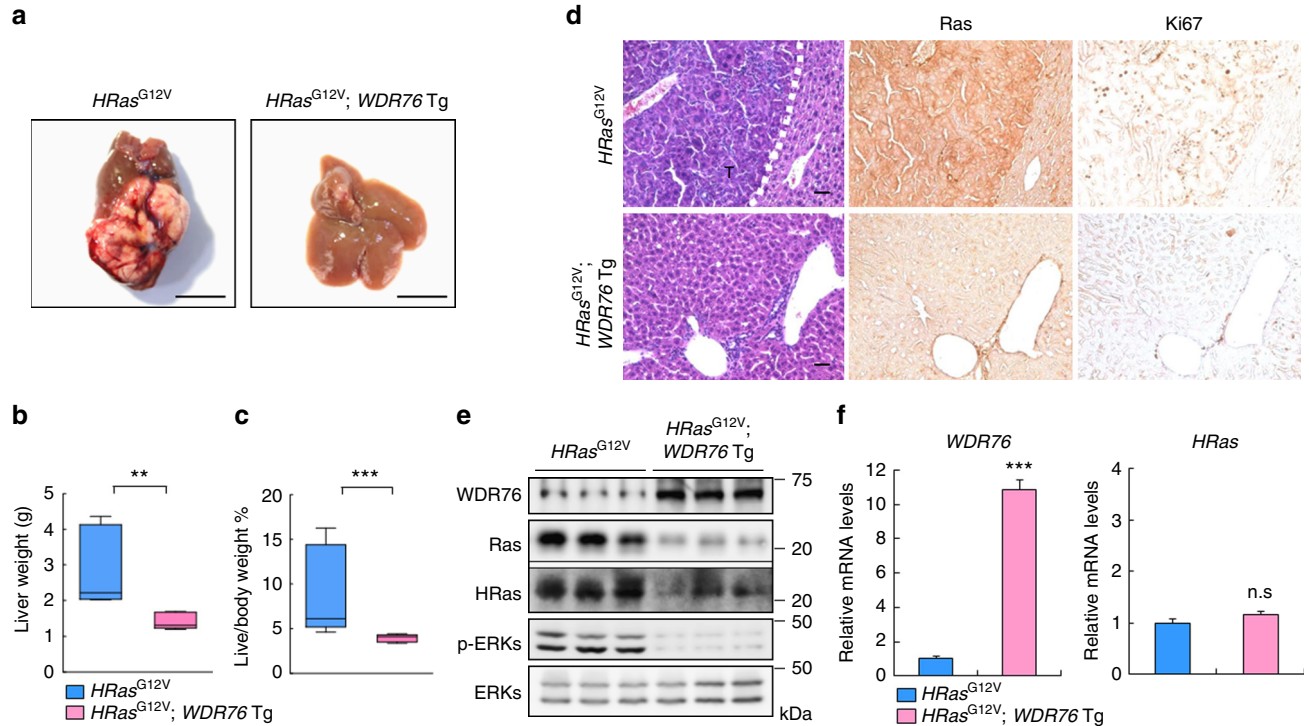

**Fig. 5** Liver-specific WDR76 inductions suppresses oncogenic HRas-driven hepatic carcinogenesis. **a** Representative gross images of livers from $HRas^{G12V}$ and $HRas^{G12V}/WDR76$ Tg mice. Scale bars, 10 mm. **b**, **c** Boxplots show liver weights (**b**), and liver-to-body weight ratios (**c**) ($n = 6$ $HRas^{G12V}$ and 6 $HRas^{G12V}/WDR76$ Tg). Data are presented as the mean ± SEM. Two-sided Student's $t$ test, ** $p < 0.01$, *** $p < 0.001$. **d**–**f** IHC (**d**), IB (**e**), and quantitative RT-PCR (**f**) analyses of liver tissues from 52-week-old $HRas^{G12V}$ and $HRas^{G12V}/WDR76$ Tg mice. Scale bars, 100 μm. **f** Data are presented as the mean ± SD ($n = 4$ biological replicates). Statistical significances were assessed using two-sided Student's $t$ test, *** $p < 0.001$

## Methods

**Mice**. All animal experiments were performed in accordance with the Korean Food and Drug Administration guidelines. Protocols were reviewed and approved by the Institutional Animal Care and Use Committee (IACUC) of Yonsei University. The ES cell clone (IST11346A1) for *WDR76* knockout were obtained from KOMP Repository and injected into blastocysts from C57BL/6 mice to derive chimera mice by Yonsei Laboratory Animal Research Center (Seoul, Korea). Plasmid DNA of pCB-WDR76 was used for generation of Cre-inducible WDR76 transgenic mice. Transgenic mice were produced, including linearization of DNA and microinjection, by Macrogen (Seoul, Korea). *WDR76* Tg mice were obtained and transgene expression was examined by crossing with albumin-CRE mouse strain (obtained from the Jackson Laboratory). $HRas^{G12V}$ mice were obtained from Consejo Superior de Investigaciones Científicas (CSIC). All mice are in the C57BL/6 background.

**DEN-induced hepatocarcinogenesis and liver injury**. For hepatocarcinogenesis, 2-week-old male $WDR76^{+/+}$ and $WDR76^{-/-}$ mice were injected intraperitoneally (i.p.) with 25 mg/kg of DEN (Sigma-Aldrich) and then sacrificed 8 months after injection for analysis of liver tumors and lung metastases. For short-term studies of inflammation and liver injury, 6-week-old male $WDR76^{+/+}$ and $WDR76^{-/-}$ mice were injected i.p. with 100 mg/kg of DEN and were sacrificed 48 h after injection.

**Measurement of the liver enzyme alanine aminotransferase**. The total blood of mice was collected by cardiac puncture. The blood was allowed to clot for 30 min and was then centrifuged for 10 min at $1000 \times g$ to obtain the supernatant. Serum levels of ALT were measured using ALT activity assays (Fujifilm, Japan), as described in the manufacturer's instructions.

**Histology, immunohistochemistry, and immunocytochemistry**. After fixation, the tissue samples were processed, embedded in paraffin, and sectioned into 4 μm slices using a RM2245 microtome (Leica Microsystems, Germany). For H&E, Sirius red, and IHC staining, the sectioned tissues were deparaffinized in xylene, hydrated in serially diluted ethanol, and stained with H&E or Sirius red solutions. For IHC, the sectioned tissues were autoclaved in 10 mM sodium citrate buffer (pH 6.0) for antigen retrieval. The tissues were treated with 0.3% hydrogen peroxide (Samchun Chemicals, Korea) to block endogenous peroxide activity before diaminobenzidine (DAB) staining. The sections were then blocked with PBS containing 5% bovine

serum albumin (BSA) and 1% normal goat serum (Vector Laboratories, CA) at room temperature for 1 h, and incubated with primary antibody overnight at 4 °C. Following primary antibodies were used for the IHC; anti-RAS (Millipore, 05-516, 1:100), anti-PCNA (Santa Cruz Biotechnology, sc-56, 1:100), anti-Ki67 (Abcam, ab15580, 1:100), and anti-αSMA (Abcam, ab7817, 1:100). For the DAB staining, the tissues were incubated with biotinylated secondary antibody (Vector Laboratories, 1:200) and avidin-biotin complex solutions (Vector Laboratories). Detection was performed with DAB substrate (Vector Laboratories), followed by Mayer's hematoxylin (Muto, Japan) counterstaining. The DAB-stained slides were visualized using a general optical microscope (TE-2000U, Nikon). TMAs composed with 46 cases of human HCC tissues with matched adjacent non-tumor tissues (LV1505) were purchased from US Biomax (Rockville, MD). Signals of the TMA slides were analyzed using a bright field microscope (Nikon TE-2000U, Japan). For the measurement of the expression levels of proteins, the intensity of each staining was quantified by IHC Profiler plugin[48]. For immunofluorescence staining, the sections were incubated with Alexa Fluor 488-conjugated IgG secondary antibody (Invitrogen, 1:200) at room temperature for 1 h, counterstained with DAPI (Boehringer Mannheim, Germany) and mounted in Gel/Mount medium. Visualization of the fluorescence signals was performed using confocal microscopy (LSM700, Carl-Zeiss) at excitation wavelengths of 488 nm (Alexa Fluor 488) and 405 nm (DAPI). SK-Hep1 cells expressing GFP-WDR76 FL, or GFP-WDR76 ΔNLS were grown on glass coverslips for the immunocytochemistry analysis. The cells were washed in PBS, fixed with 4% paraformaldehyde, permeabilized with 0.1% Triton X-100, and blocked with 5% BSA. GFP-tagged WDR76 and RAS proteins were detected using anti-GFP (Santa Cruz Biotechnology, sc-8334, 1:200) or anti-RAS (Millipore, 05-516, 1:100) antibody followed by Alexa Fluor 555- or Alexa Fluor 488-conjugated IgG secondary antibody (Invitrogen, 1:200).

**Cell culture, transfection, and reagents**. Human liver cancer cells (SK-Hep1, Huh7, HepG2, PLC/PRF/5, and Hep3B), Lovo (CRC), T24T (bladder cancer), human embryonic kidney 293 (HEK 293), and HEK293T cells were obtained from the American Type Culture Collection (Manassas, VA). $WDR76^{+/+}$ or $WDR76^{-/-}$ MEF cells were prepared from E13.5 mouse embryos. SK-Hep1, Huh7, HepG2, Lovo, T24T, HEK 293, HEK293T, and MEF cells were grown in Dulbecco's modified Eagle's medium (DMEM) (Gibco), and PLC/PRF/5 and Hep3B cells were grown in RPMI-1640 medium (Gibco). All culture media were supplemented with 10% (v/v) fetal bovine serum (Gibco) and 100 U/ml streptomycin and penicillin (Gibco). The MEF cells were also supplemented with 2 mM L-glutamine (Gibco).

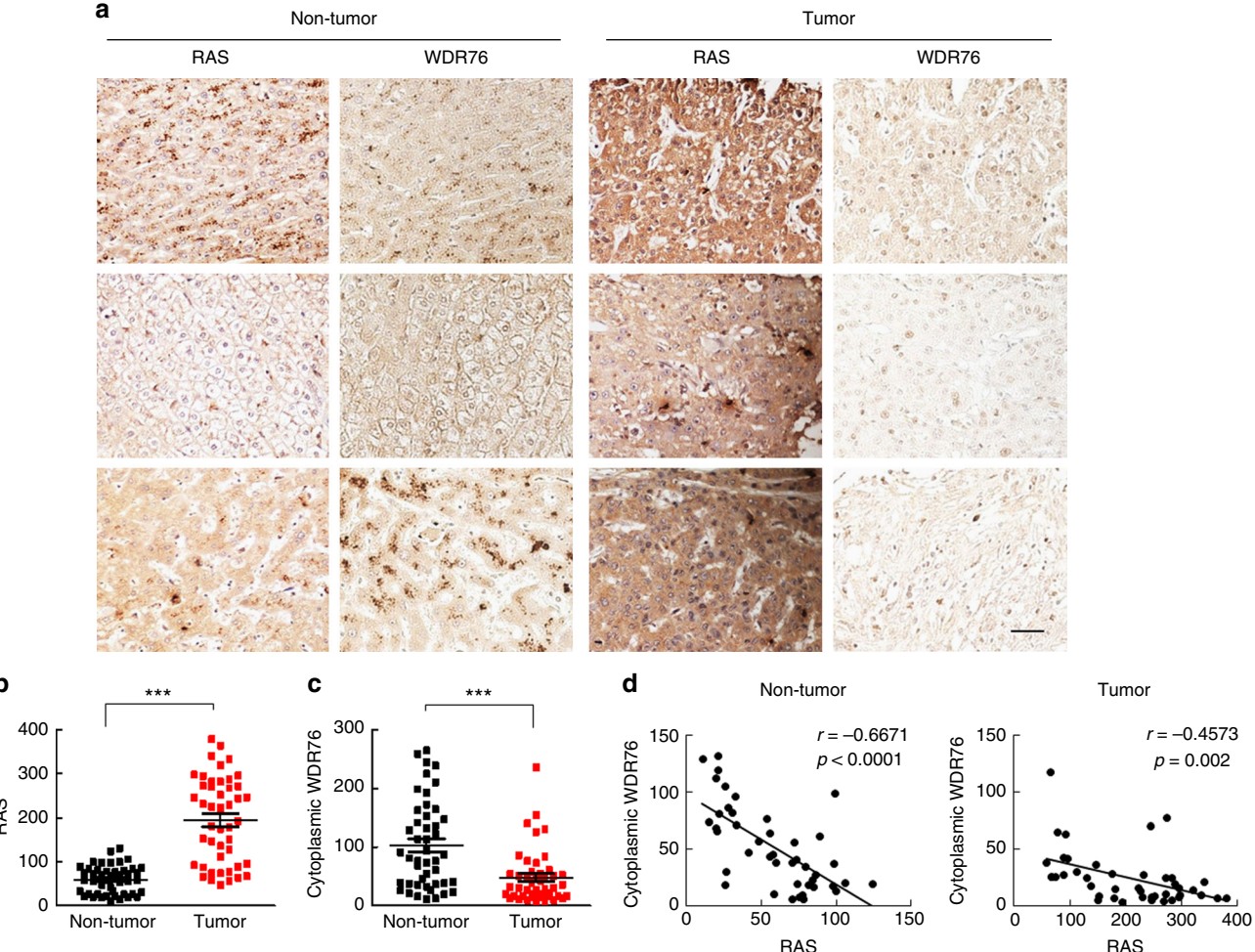

**Fig. 6** Expression of RAS and WDR76 proteins in tumor and adjacent non-tumor tissues from human HCC. **a** Representative DAB staining images of RAS and WDR76 from tumor and paired adjacent non-tumor tissues fixed on human HCC TMA slides. **b**, **c** Quantitative analyses of RAS (**b**) and cytoplasmic WDR76 (**c**) were performed by comparing tumor and adjacent non-tumor tissues based on H-Score. **d** A graph showing the inverse correlation between RAS and WDR76 expression of 46 sets of human HCC tissues with matched adjacent non-tumor tissues. All values were calculated using Student's $t$ test or Pearson correlation analyses. ***$p < 0.001$. Scale bars, 100 μm

All cells were cultured at 37 °C in a humidified 5% $CO_2$ incubator. The cells were transiently transfected with lipofectamine (Invitrogen), following the manufacturer's instructions. The following reagents were administered at the indicated concentrations: CHX (50 μg/ml; Sigma-Aldrich), ALLN (25 μg/ml; Sigma-Aldrich), and TGFβ (5 ng/ml; Peprotech, NJ).

**Lentivirus production and stable cell line generation**. Virus productions were carried out using HEK293T cells. Briefly, the cells were transfected with lentiviral DNA constructs together with the viral packaging psPAX2 and viral envelope pMD2G plasmids at a 2:1:1 ratio, respectively. The virus supernatants were harvested at 24 and 48 h post-transfection, filtered (0.3-μm pore size), and used for the infections.

The *WDR76*-knockout SK-Hep1 cells (SK-KO) were generated using CRISPR/Cas9 method[49]. The guide sequences targeting Exon 5 of *WDR76* (5′-GCTGGTAATGAAACTCCC-3′) were cloned into the plasmid lentiCRISPRv2 (Addgene) for virus production. To obtain single-cell clones for stable cell line generation, SK-Hep1 cells were transduced with the WDR76 KO lentivirus and selected with puromycin (Sigma-Aldrich). To establish WDR76 rescue cell lines, SK-KO cells were transduced with GFP, GFP-WDR76 FL, or GFP-WDR76 ΔNLS lentivirus and were selected with Hygromycin B (Duchefa, The Netherlands).

**Immunoprecipitation, immunoblotting, and ubiquitination assays**. Immunoprecipitation (IP), immunoblotting (IB), GST pull-down, and ubiquitination assays were performed as previously described[7]. Briefly, cell lysates prepared using ice-cold radioimmunoprecipitation assay (RIPA) buffer (Millipore) was applied to IP or IB assays, with the appropriate antibodies. For the ubiquitinayltion assay in cells, 10 mM N-ethylmaleimide (NEM; Sigma-Aldrich) was subsequently added to the

RIPA buffer. The cell lysates were subjected to IP with the indicated antibodies. The ubiquitin-conjugated proteins were detected by IB. For the ubiquitination assays under denaturing conditions, denatured cell extracts were prepared by resuspending cell pellets in 1 ml of denaturing buffer [50 mM tris (pH 7.5), 150 mM NaCl, 1% SDS, and 10 mM NEM] and boiled for 10 min. IPs were performed with an antibody recognizing Myc after addition of 9 ml of tris-buffered saline (TBS) buffer [50 mM tris (pH 7.5) and 150 mM NaCl] together with 0.5% NP-40 and 10 mM NEM. To detect the ubiquitinated RAS proteins, we captured the ubiquitinated proteins using an UbiQapture-Q Kit[50] (Enzo Life Sciences, Switzerland) according to the manufacturer's instructions. Cells were collected and lysed with RIPA buffer. We then added 50 μl of UbiQapture-Q matrix to the cell lysate. The sample was resuspended gently by inversion at 4 °C overnight to allow the ubiquitinated protein conjugates to bind to the affinity matrix. After centrifugation at 5000 × $g$ for 30 s, the matrix was washed twice in phosphate-buffered saline (PBS). Extracted ubiquitinated proteins were then subjected to immunoblotting using anti-RAS (Millipore, 05-516, 1:3000) antibody. For the GST pull-down assay, bacterially expressed GST-H, K, or NRAS were purified using glutathione agarose beads (BD Biosciences). The cell lysates were incubated with the purified soluble GST fusion proteins and the pull-down samples were subjected to IB analyseis. Following primary antibodies were used for IB; anti-Ras (Millipore, 05-516, 1:3000), anti-HRas (Santa Cruz Biotechnology, sc-520, 1:500), anti-KRas (Santa Cruz Biotechnology, sc-30, 1:500), anti-NRas (Santa Cruz Biotechnology, sc-31, 1:500), anti-p-ERK (Cell Signaling Technology, #9101S, 1:1000), anti-ERK (Santa Cruz Biotechnology, sc-514302, 1:5000), anti-PCNA (Santa Cruz Biotechnology, sc-56, 1:3000), anti-N-cadherin (BD Bioscience, #610920, 1:3000), anti-αSMA (Abcam, ab7817, 1:3000), anti-Myc (Cell Signaling Technology, #2276S, 1:3000), anti-FLAG (Sigma-Aldrich, F7425, 1:3000), anti-HA (Santa Cruz Biotechnology, sc-7392, 1:3000), anti-V5 (MBL International., M167-3, 1:3000), anti-

GFP (Santa Cruz Biotechnology, sc-8334, 1:3000), anti-GST (Santa Cruz Biotechnology, sc-374171, 1:3000) and anti-β-actin (Santa Cruz Biotechnology, sc-47778, 1:3000). WDR76 polyclonal antibody was generated from immunization of rabbits with partially purified WDR76 proteins (GST-WDR76 1-300; Abfrontier, Korea). The antibody was purified using ProteinA-Sepharose and a standard procedure. Secondary antibodies, horseradish peroxidase-conjugated anti-mouse antibody (Cell Signaling Technology, #7076, 1:5000) or anti-rabbit antibody (Bio-Rad, #1706515, 1:5000) were used in this study. The band signals were acquired with a LAS-4000 LCD camera coupled to MultiGauge software (Fuji). Uncropped blots are available in Supplementary Fig. 8–14.

**Identification of HRAS binding proteins**. To minimize nonspecific protein binding, the tissue extracts were pre-cleared with glutathione agarose bead (BD Biosciences). The tissue lysates were incubated with purified GST-HRAS or GST overnight at 4 °C. Glutathione agarose beads were then added, and incubation continued for 2 h at 4 °C. The GST-HRAS pulled-down agarose beads were washed three times with RIPA buffer and were boiled in the SDS sample buffer. The multiprotein complexes obtained by pull-down experiments were separated on a SDS-PAGE gel followed by Coomassie blue staining. The resulting bands were extracted from the gel and subjected to LC-MS/MS sequencing and data analysis. In brief, each stained gel was divided into 22 slices, and the proteins in the gel were digested and extracted. Separation and analysis of the tryptic peptides were performed using a nano LC-ESI-MS/MS system, combining an Ultimate nano LC systems including the FAMOS autosampler and Switchos column switching valve (LC-Packings, The Netherlands) connected to a QSTAR mass spectrometer (Applied Biosystems, CA) with a nanospray interface[51]. The obtained spectra were automatically processed and compared with the NCBI non-redundant database using the MASCOT software package (Matrix Sciences, UK).

**Cell proliferation, colony formation, and invasion assays**. For cell proliferation assay, the cells were seeded into a 96-well plate and viable cell numbers were determined at 0 and 96 h after seeding. After a 2-h incubation in 3-(4,5-dimethylthiazol-2-yl)-2-5-diphenyltetrazolium bromide (MTT; AMRESCO) reagent at 37 °C, 200 μl of dimethyl sulfoxide (per well) was added to dissolve formazan crystals, and the optical density (590-nm wavelength) was measured using a FLUOstar OPTIMA microplate reader (BMG LABTECH). For the BrdU (5-bromo-2′-deoxyuridine) incorporation assay, the cells were grown on glass coverslips and pulsed for 3 h with BrdU (Sigma-Aldrich) before harvest; they were then immunostained with an anti-BrdU antibody (Dako, M0744). For the colony formation assays, the cells were seeded into 12-well plates (100–250 cells/well) and cultured for 14–21 days. At the end of the experiments, cells were stained with 0.5% crystal violet in 20% ethanol, and then washed three times with distilled water. For the invasion assays, the cells were seeded onto Matrigel-coated chambers and were allowed to invade for 18 h. After clearing the cells from the inner surface of the chamber, the cells on the outer surface were fixed with 4% paraformaldehyde (PFA) and stained with crystal violet. Each chamber was dipped in distilled water to remove the excess stain and allowed to dry. The photographic images were acquired using a general optical microscope (TE-2000U, Nikon).

**Live cell imaging**. SK-WT-GFP, SK-KO-GFP, SK-KO-GFP-WDR76 FL, or SK-KO-GFP-WDR76 ΔNLS cells were plated onto fibronectin-coated chambers (Nunc Lab-Tek, MA). Simultaneous acquisition of two colors was performed using a Nikon Eclipse Ti microscope (Nikon) equipped with a Chamlide TC incubator system (Live Cell Instrument), which maintained 37 °C and 5% CO$_2$ humidity conditions during live cell imaging. Photographic images were taken every 20 min for 24 h. Movies were made using NIS viewer software (Nikon).

**Human liver samples**. Primary human HCC patient tumor tissues and corresponding adjacent normal tissues were obtained in accordance with research ethics board approval from Severance Hospital, Yonsei University (Seoul, Korea). Informed consent was obtained from each patient. All the samples taken after surgery were stored in liquid nitrogen for immunoblotting assay.

**RNA analysis**. Total RNAs were extracted from tissues or cells using TRIzol reagent (Invitrogen) according to the manufacturer's instructions. The cDNA was synthesized with M-MLV reverse transcriptase (Invitrogen). The PCR reactions were performed with Ex-Taq DNA polymerase (SG bio, Korea). Real-time quantitative PCR was performed using iQ SYBR Green Supermix (Bio-Rad). The relative changes in gene expression were measured using the comparative cycle-threshold (CT) method; the results were normalized to the geometric mean of GAPDH, β-actin, and HPRT. Primer sequences are listed in Supplementary Table 2.

**In situ hybridization**. In situ hybridization was performed as previously described[44]. Paraformaldehyde-fixed paraffin sections were deparaffinized and rehydrated. Sections were then incubated in fixation solution (4% PFA in PBS) at room temperature for 10 min. To remove protein, the sections were incubated in proteinase K solution (10 μg/ml in 50 mM Tris-HCl and 5 mM EDTA, pH 8.0) for 10

min at 37 °C, incubated in 5X SSC buffer for 15 min. The sections were then incubated with pre-hybridization buffer (5X SSC, 50% formamide, pH to 7.5 with HCl, 0.1% Tween-20, 50 μg/ml salmon sperm DNA, 50 μg/ml yeast tRNA) for 2 h at 58 °C. After this step, digoxigenin-riboprobes (100–500 ng/ml) were hybridized in the pre-hybridization buffer for 24 h at 58 °C. The sections were then rinsed with 2X SSC and 0.1X SSC buffer for 1 h at 65 °C. They were then washed with PBS and incubated in PBS containing 1% BSA and 0.2% Triton X-100 for 30 min. After this step, they were incubated with anti-digoxigenin-AP antibody (Roche) diluted 1:500 in PBS containing 1% BSA and 1% NGS for 1 h followed by washing the slides three times with PBS for 5 min each. The sections were then incubated in NBT/BCIP solution (Sigma-Aldrich) to achieve colorization.

**Statistical analysis**. The results were expressed as mean ± standard deviation (s.d.) values of at least triplicate experiments. The statistical significance of differences was assessed using the Student's $t$ test or Pearson correlation analyses. Significance for Pearson correlation coefficient ($r$) was determined by $p$-value. $p < 0.05$ was considered a statistically significant difference (*$p < 0.05$, **$p < 0.01$, and ***$p < 0.001$). The statistical computations were performed using Prism software (Graph Pad).

**Reporting Summary**. Further information on experimental design is available in the Nature Research Reporting Summary linked to this article.

## Data availability
The data that support the findings of this study are available within this article and its supplementary information and/or from the corresponding author upon reasonable request.

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

## Acknowledgements

This work was supported by the National Research Foundation of Korea (NRF) grant funded by the Korean Government (MSIP) (grants 2016R1A5A1004694, 2015R1A2A1A05001873).

## Author contributions

W.J.J., J.C.P., E.J.R., and S.H.J. performed the experiments. W.S.K. performed LC-MS-MS analysis. Y.N.P. provided well-characterized patient samples. W.J.J., J.C.P., E.J.R., S.K.L., D.S.N., and K.Y.C. performed data analysis and wrote the manuscript.

## Additional information

**Competing interests:** The authors declare no competing interests.

