## [Peer Review File · Nature Communications]

Reviewers' Comments:

Reviewer #1:

Remarks to the Author:

This study described a thus far unknown link between WDR76 and RAS oncogene activity thus contributing to liver carcinogenesis. Starting with the MS experiment, the authors revealed that H-RAS interacts with WDR76 only in normal tissue. The authors then employed overexpression studies in cell culture models to suggest that WDR76 is possibly contributing to ubiquitination of RAS in a CUL4a dependent manner. Further studies in a mouse model show a link between WDR76, RAS-MAPK signaling, and liver carcinogenesis. The study despite revealing an important link suffers from clear mechanistic insights and pathological relevance.

Major concerns:

1. The relevance of the proposed mechanism to human liver cancer remains unclear. The authors suggested that WDR76 acts as a tumor suppressor in hepatocarcinoma. However, no WDR76 mutations or deletions have been identified in the TCGA hepatocarcinoma cohort. Heterozygous loss of WDR76 is observed in about 20% of liver cancers, but this loss does not lead to WDR76 down-regulation, making doubtful a potential role of WDR76 in tumor suppression. There is also no indication that the MS experiments provided a deep coverage of the RAS interactome. The interaction between WDR76 and RAS could be missed in cancer cells due to the coverage issues. The authors should provide an additional proof for the loss of interaction between RAS and WDR76 in hepatocarcinomas. They should also explain why RAS is losing the ability to interact with WDR76 in cancer.
2. Why did the author exclude several components of the ubiquitin system (such as USP29, ZBTB10, and UBE3B), which were identified by the MS experiments, from further validations?
3. The data describing the link to ubiquitination of RAS with Cul4a needs further experimentation and strengthening. The complex formation between WDR76/CUL4a and RAS is not shown.
4. The ubiquitination experiments presented should also be performed under denaturing conditions to prove that ubiquitin covalently conjugated to RAS.
5. RAS ubiquitination is shown only in ubiquitin overexpressing cells, whereas multiple studies demonstrated that ubiquitin affects the physiology and bioactivity of the ubiquitin molecule (reviewed in (Emmerich and Cohen 2015)).
6. Fig. 2C. It is unclear from the legend how RAS localization was analyzed. Did the author use pan-RAS antibody? Why did they observe RAS in the nucleus? Moreover, the cytoplasmic localization of WDR76 looks like the artefact of extremely high overexpression in a particular cell. In this cell, RAS is also co-localized with WDR76 also in the nucleus. The better images should be provided. Ideally, this experiment should be also done with endogenous WDR76, especially taking into the majority of WDR76 is localized in the nucleus.
7. The biological relevance of the WDR76-NLS mutant is unclear.
8. If WDR76 loss drives tumorigenic transformation via activation of the MAPK signaling, does the tumorigenic phenotype reversed in the presence of RAF or MEK inhibitors.
9. Does CUL4A loss phenocopy WDR76 loss phenotype in hepatocarcinoma development?
10. Previous studies describing RAS degradation mechanisms should be discussed.

Reviewer #2:

Remarks to the Author:

In the present manuscript, the authors investigated the molecular mechanisms responsible for unrestrained activity of Ras in cancer. The authors identified WD40-repeat 9 protein 76 (WDR76) as one of new H-RAS binding proteins using proteomic analyses of human hepatocellular carcinomas (HCC) specimens. WDR76-dependent destabilization of Ras triggered inhibition of proliferation, transformation, and invasion of HCC cells. In addition, WDR76 knockout mice were more susceptible to diethylnitrosamine-induced liver carcinogenesis. Conversely, liver-specific

WDR76 induction destabilized Ras and significantly decreased hepatocarcinogenesis. The authors conclude that WDR76 functions acts as a tumor suppressor via mediating degradation. This is an excellent manuscript, providing novel and important findings in the cancer field. The data are solid and fully support the conclusions drawn. The experiments were properly planned and the methodology used was highly appropriate. Figures are easy to understand as well. Some issues have to be addressed to further increase the value of the present manuscript.

Major issues:

1. To definitively prove the tumor suppressive role of WDR76, the authors should evaluate its levels in a collection of human HCC specimens and respective non-tumorous surrounding tissues or, alternatively, in another tumor type.
2. The HepG2 and SK-Hep1 cell are not HCC cell lines, but hepatoblastoma and liver adenocarcinoma cell lines, respectively. Thus, the authors should indicate these cell lines throughout the text as "liver cancer cell lines".

Minor issues:

1. Either in the Introduction or Discussion section of the manuscript, the authors should describe the other mechanisms besides mutations involved in Ras unconstrained activity in HCC, including inactivation of Ras GAPs, Spry2, Dusp1, etc. tumor suppressors, and include the appropriate references.

Point-by-point response to the reviewers

We appreciate the reviewers for their constructive comments and have used the comments to revise the manuscript. The manuscript is substantially improved by performing new experiments to address the referee's critiques. Moreover, we re-formatted the manuscript as the editor recommended to fit the guidelines provided by *Nature Communications*.

We hope that our revised manuscript is now suitable for publication in *Nature Communications*.

Reviewers' comments:

Reviewer #1 (Remarks to the Author):

This study described a thus far unknown link between WDR76 and RAS oncogene activity thus contributing to liver carcinogenesis. Starting with the MS experiment, the authors revealed that H-RAS interacts with WDR76 only in normal tissue. The authors then employed overexpression studies in cell culture models to suggest that WDR76 is possibly contributing to ubiquitination of RAS in a CUL4a dependent manner. Further studies in a mouse model show a link between WDR76, RAS-MAPK signaling, and liver carcinogenesis. The study despite revealing an important link suffers from clear mechanistic insights and pathological relevance.

Major concerns:

1. The relevance of the proposed mechanism to human liver cancer remains unclear. The authors suggested that WDR76 acts as a tumor suppressor in hepatocarcinoma. However, no WDR76 mutations or deletions have been identified in the TCGA hepatocarcinoma cohort. Heterozygous loss of WDR76 is observed in about 20% of liver cancers, but this loss does not lead to WDR76 down-regulation, making doubtful a potential role of WDR76 in tumor suppression. There is also no indication that the MS experiments provided a deep coverage of the RAS interactome. The interaction between WDR76 and RAS could be missed in cancer cells due to the coverage issues. The authors

should provide an additional proof for the loss of interaction between RAS and WDR76 in hepatocarcinomas. They should also explain why RAS is losing the ability to interact with WDR76 in cancer.

Response: We first appreciate the reviewer for the important critiques addressing pathological relevance of our mechanism of RAS degradation via WDR76. Protein stability of E3 ubiquitin ligases that endow the system for specificity towards many different substrates is also regulated by the proteasome-dependent mechanisms¹⁻⁴.

Although mutations, deletions, or down-regulation of the WDR76 gene may not have been frequently observed in liver cancers as reviewer commented, the E3 ligase WDR76 could possibly be regulated at the protein level under different pathophysiological conditions.

To validate the possible tumor suppressive role of WDR76 in human hepatocellular carcinoma (HCC), we analyzed a tissue microarray (TMA; LV1505; US Biomax) consisting of 46 cases of human HCC tissues with paired adjacent non-tumor tissues. As shown by immunohistochemical analyses, we found the RAS staining intensity was significantly higher in tumor tissues compared with paired non-tumor tissues, and WDR76 staining pattern was inversely correlated in the HCC tissues. Especially, expression levels of WDR76 were significantly lower in HCC patient tumors compared with matched adjacent non-tumor tissues (Fig. 6). This inverse correlation of RAS and WDR76 is consistent with our current study identifying the RAS degradation via WDR76. Moreover, cytoplasmic localization of WDR76 and its lowered levels in tumor tissues compared with paired non-tumor tissues supports our data providing a mechanism for the RAS degradation by WDR76 in the cytoplasm of liver cells.

Although we understand the reviewer's concern that the interaction between WDR76 and RAS could be missed in cancer cells due to the coverage issues, the overall data propose that low expression levels of cytoplasmic WDR76 protein may have lowered the amount of its interaction with RAS in liver cancer. Taken together, these results indicate that the tumor suppressive role of WDR76,

especially cytosolic WDR76, in liver cancer mainly functions through modulation of the stability regulation of RAS.

2. Why did the author exclude several components of the ubiquitin system (such as USP29, ZBTB10, and UBE3B), which were identified by the MS experiments, from further validations?

Response: As suggested by the reviewer, we newly performed knock down experiments with addition of USP29 and UBE3B which are indicated by the reviewer except ZBTB10 (Zinc Finger and BTB Domain Containing 10). The ZBTB10 is not described as a component of the ubiquitin system, but as a protein mainly involved in transcriptional regulation. In the experiments, we confirmed the specific up-regulation of the RAS by the siRNA mediated knock down of WDR76, but not by that of HectD1, UBE4B, USP42, UBE3B or USP29. We replaced the data as shown in Fig. 1f (Supplementary Fig. 1d of the 1st draft) in the revised manuscript.

3. The data describing the link to ubiquitination of RAS with Cul4a needs further experimentation and strengthening. The complex formation between WDR76/CUL4a and RAS is not shown.

Response: As requested by the reviewer, we conducted immunoprecipitation assays in HEK293 cells transfected with H-RAS, WDR76, and CUL4A. We then confirmed complex formation between WDR76/CUL4A and RAS by immunoprecipitation, and the data were newly provided as Supplementary Fig. 1i in the revised manuscript.

4. The ubiquitination experiments presented should also be performed under denaturing conditions to prove that ubiquitin covalently conjugated to RAS.

Response: To resolve the reviewer's concern, we performed RAS ubiquitination assay under

denaturing conditions and confirmed the polyubiquitination of RAS by WDR76 overexpression in the denaturing condition. The data are presented in Supplementary Fig. 1g.

5. RAS ubiquitination is shown only in ubiquitin overexpressing cells, whereas multiple studies demonstrated that ubiquitin affects the physiology and bioactivity of the ubiquitin molecule (reviewed in (Emmerich and Cohen 2015)).

Response: We appreciated the professional suggestions by the reviewer. To address this concern, we newly performed WDR76-mediated RAS ubiquitination assays using UbiQapture-Q Matrix and captured ubiquitinated RAS proteins using an anti-RAS antibody. The enrichment of polyubiquitinated RAS proteins by overexpression of WDR76 was confirmed as shown in Supplementary Fig. 1h. These data confirm that WDR76 mediates the RAS ubiquitination.

6. Fig. 2C. It is unclear from the legend how RAS localization was analyzed. Did the author use pan-RAS antibody? Why did they observe RAS in the nucleus? Moreover, the cytoplasmic localization of WDR76 looks like the artefact of extremely high overexpression in a particular cell. In this cell, RAS is also co-localized with WDR76 also in the nucleus. The better images should be provided. Ideally, this experiment should be also done with endogenous WDR76, especially taking into the majority of WDR76 is localized in the nucleus.

Response: The authors thank the reviewer for these critiques to clarify data. In Fig. 2c (1st draft of the manuscript), to evaluate the role of cytoplasmic WDR76 on enhanced RAS destabilization, we previously transfected SK-Hep 1 cells with either GFP-WDR76FL or GFP-WDR76ΔNLS, and compared immunocytochemical analyses data obtained by using the anti-RAS or anti-GFP antibody. However, the qualities of images obtained from these transient transfection experiments were poor, and it could mislead the data interpretation as the reviewer commented.

To provide better images, we newly performed immunocytochemical analyses with SK-HEP1 stable cell lines overexpressing GFP-WDR76FL or GFP-WDR76ΔNLS (these are the same cell lines that have been used for the evaluation of the proliferation and transformation of cells as shown in Supplementary Fig. 2d-f of the revised manuscript). In these data, we confirm the role of cytoplasmic WDR76 in the degradation of RAS by providing the images clearly visualizing GFP and RAS signals. The Fig. 2c of the 1st draft is now replaced by the new images as shown in Fig. 3c of the revised manuscript.

7. The biological relevance of the WDR76-NLS mutant is unclear.

Response: We only used the construct in the experiments to distinguish WDR76's roles dependent upon its subcellular localization (cytoplasm vs nucleus) in the degradation of RAS. The experiments using WDR76ΔNLS construct supports our study showing the biological role of cytoplasmic WDR76 in the degradation of RAS and the role as a tumor suppressor.

8. If WDR76 loss drives tumorigenic transformation via activation of the MAPK signaling, does the tumorigenic phenotype reversed in the presence of RAF or MEK inhibitors.

Response: To address this question, we assessed whether WDR76 KO-induced cell transformation occurs via activation of the MAPK pathway. The AS703026, a MEK inhibitor, suppressed WDR76 KO-induced transformation of the SK-KO cells, indicating that WDR76 KO-mediated RAS stabilization induces cell transformation through the MAPK pathway. The data are presented in Supplementary Fig. 3f.

9. Does CUL4A loss phenocopy WDR76 loss phenotype in hepatocarcinoma development?

Response: Because CUL4A is known to use WDR proteins as molecular adaptors for substrate recognition and WDR76 is identified as Cul4A interacting protein^{5, 6}, we validated the effect of CUL4A as a component of the CUL4-DDB1-WDR76 ubiquitin ligase complex⁵ on stability regulation of RAS.

The CUL4A ubiquitin ligase complexes which promote the ubiquitination of a variety of substrates play roles in a wide range of cellular processes including cell cycle, signaling, development, DNA damage response, and chromatin remodeling⁷. There are reports that CUL4A has been shown to target proto-oncogenic targets such as N- and C-Myc and c-Jun^{8, 9}. However, there are also reports about the association with tumor suppressor genes such as p53 and MDM2¹⁰. Therefore roles of CUL4A in cancer development could be highly complicated depend on diverse physiological contexts.

10. Previous studies describing RAS degradation mechanisms should be discussed.

Response: As the reviewer suggested, we discussed previous studies illustrated RAS degradation mechanisms with appropriate citations in the Introduction and Discussion sections in the revised manuscript.

Reviewer #2 (Remarks to the Author):

In the present manuscript, the authors investigated the molecular mechanisms responsible for unrestrained activity of Ras in cancer. The authors identified WD40-repeat 9 protein 76 (WDR76) as one of new H-RAS binding proteins using proteomic analyses of human hepatocellular carcinomas (HCC) specimens. WDR76-dependent destabilization of Ras triggered inhibition of proliferation, transformation, and invasion of HCC cells. In addition, WDR76 knockout mice were more susceptible to diethylnitrosamine-induced liver carcinogenesis. Conversely, liver-specific WDR76 induction destabilized Ras and significantly decreased hepatocarcinogenesis. The authors conclude that WDR76

functions acts as a tumor suppressor via mediating degradation.

This is an excellent manuscript, providing novel and important findings in the cancer field. The data are solid and fully support the conclusions drawn. The experiments were properly planned and the methodology used was highly appropriate. Figures are easy to understand as well. Some issues have to be addressed to further increase the value of the present manuscript.

Major issues:

1. To definitively prove the tumor suppressive role of WDR76, the authors should evaluate its levels in a collection of human HCC specimens and respective non-tumorous surrounding tissues or, alternatively, in another tumor type.

Response: We appreciate the reviewer for this important suggestion. To validate the possible tumor suppressive role of WDR76 in human HCC, we analyzed a tissue microarray (TMA; LV1505; US Biomax) consisting of 46 cases of human hepatocellular carcinoma with paired adjacent non-tumor tissue samples. IHC analyses of these tissues revealed that RAS levels were higher in 41/46 (89.1%) HCC patient tumors compared with their adjacent non-tumor tissue, whereas WDR76 expression appeared lower in 34/46 (73.9%) of HCC tumors as shown in Fig. 6. Comparison of the protein expression ratios (tumor/non-tumor) of RAS and WDR76 from the regions of the tumor and paired non-tumor tissues further revealed a significant negative correlation between the levels of RAS and WDR76 (non-tumor; $r = -0.6671$, $p < 0.001$ and tumor; $r = -0.4573$, $p = 0.002$) (Fig. 6). Thus, there is a significant association between RAS and WDR76 levels in human HCCs.

2. The HepG2 and SK-Hep1 cell are not HCC cell lines, but hepatoblastoma and liver adenocarcinoma cell lines, respectively. Thus, the authors should indicate these cell lines throughout the text as "liver cancer cell lines".

Response: We appreciated the direction by the reviewer. We indicated HepG2 and SK-Hep1 cell lines as liver cancer cell lines throughout the text in the revised manuscript.

Minor issues:

1. Either in the Introduction or Discussion section of the manuscript, the authors should describe the other mechanisms besides mutations involved in Ras unconstrained activity in HCC, including inactivation of Ras GAPs, Spry2, Dusp1, etc. tumor suppressors, and include the appropriate references.

Response: As directed by the reviewer, we added the description of the other mechanisms besides mutations involved in RAS unconstrained activity in HCC, including inactivation of RAS GAPs, Spry2, Dusp1 tumor suppressors in the Discussion section with citations of appropriate references in the revised manuscript.

1. Inuzuka, H. et al. Phosphorylation by casein kinase I promotes the turnover of the Mdm2 oncoprotein via the SCF(beta-TRCP) ubiquitin ligase. *Cancer Cell* **18**, 147-159 (2010).
2. Linares, L.K. et al. Intrinsic ubiquitination activity of PCAF controls the stability of the oncoprotein Hdm2. *Nat Cell Biol* **9**, 331-338 (2007).
3. Xu, C., Fan, C.D. & Wang, X. Regulation of Mdm2 protein stability and the p53 response by NEDD4-1 E3 ligase. *Oncogene* **34**, 281-289 (2015).
4. de Bie, P. & Ciechanover, A. Ubiquitination of E3 ligases: self-regulation of the ubiquitin system via proteolytic and non-proteolytic mechanisms. *Cell Death Differ* **18**, 1393-1402 (2011).
5. Higa, L.A. et al. CUL4-DDB1 ubiquitin ligase interacts with multiple WD40-repeat proteins and regulates histone methylation. *Nat Cell Biol* **8**, 1277-1283 (2006).
6. Higa, L.A. & Zhang, H. Stealing the spotlight: CUL4-DDB1 ubiquitin ligase docks WD40-

- repeat proteins to destroy. *Cell Div* **2**, 5 (2007).
7. Sharma, P. & Nag, A. CUL4A ubiquitin ligase: a promising drug target for cancer and other human diseases. *Open Biol* **4**, 130217 (2014).
 8. Choi, S.H., Wright, J.B., Gerber, S.A. & Cole, M.D. Myc protein is stabilized by suppression of a novel E3 ligase complex in cancer cells. *Genes Dev* **24**, 1236-1241 (2010).
 9. Wertz, I.E. et al. Human De-etiolated-1 regulates c-Jun by assembling a CUL4A ubiquitin ligase. *Science* **303**, 1371-1374 (2004).
 10. Banks, D. et al. L2DTL/CDT2 and PCNA interact with p53 and regulate p53 polyubiquitination and protein stability through MDM2 and CUL4A/DDB1 complexes. *Cell Cycle* **5**, 1719-1729 (2006).

Reviewers' Comments:

Reviewer #1:

Remarks to the Author:

Even though the authors made a nice progress in validating their conclusions, not all technical issues are resolved and some important controls are missed.

Revised Figure 3c: the quality/magnification of the images does not allow me to make any conclusion about potential co-localization of WDR76 and RAS.

Whereas the pattern of ubiquitination shown in Suppl. Fig. 1g is typical for poly-ubiquitination/degradation, the pattern of RAS ubiquitination observed in UbiQapture Matrix does not look like poly-ubiquitination. Could author provide an explanation for this? How does suppression of WDR76 affect the levels of ubiquitination of endogenous RAS?

Suppl. Fig. 1i: Why does RAS have different molecular weights in the lysate and after pull-down?

Fig. 2d and Fig.5f : It is also not appropriate to use a single housekeeping gene as an internal control. Normally, it should be a geometric mean of several housekeeping genes.

Fig. 4e: I am surprised that there is no variation in Ki67-staining of WDR76-positive liver tumours. I am wondering whether all tumours were exactly the same?

Suppl. Fig.3f: the key control of SK-wt cells treated with the MEK inhibitor is missed. If the tumorigenic effect of WDR76 loss is MAPK-dependent, MEK inhibition should abolish the difference between wt- and WDR-KO cells.

Minor points:

1. Fig.2f,g: molecular weight markers should be added
2. Fig. 2d: n? what is the bar for? SD or SEM? T-test?
3. Fig. 2e: n? SD or SEM? Is it significant by two-way ANOVA?
4. Fig. 3f: molecular weight markers should be added
5. Fig. 4c: what does the middle line show? SD or SEM? Statistics?
6. Fig. 4e: number of analysed tumours; What do the bars show? SD or SEM? Statistics?
7. Fig.5 b,c: what does the middle line show? SD or SEM? Statistics?
8. Fig. 5f: number of mice? What do the bars show? SD or SEM? Statistics?
9. Suppl. Fig. 1g, h: molecular weight markers should be added
10. NEDD4-mediated degradation of RAS and its role in cancer development is not mentioned <https://www.ncbi.nlm.nih.gov/pubmed/24746824>

Reviewer #2:

Remarks to the Author:

The authors have satisfactorily addressed the concerns raised by the reviewer. This is an excellent study that will attract the interest of a large audience of scientists.

Point-by-point response to the reviewer

We appreciate the reviewer for his or her detailed guide and suggestions for improvement of the manuscript. The manuscript is substantially improved by performing new experiments in response to the referee's critiques. Moreover, we re-formatted the manuscript as the editor recommended to fit the guidelines provided by *Nature Communications*.

We hope that our revised manuscript is now suitable for publication in *Nature Communications*.

Reviewers' comments:

Reviewer #1 (Remarks to the Author):

Even though the authors made a nice progress in validating their conclusions, not all technical issues are resolved and some important controls are missed.

Revised Figure 3c: the quality/magnification of the images does not allow me to make any conclusion about potential co-localization of WDR76 and RAS.

Response: The goal of experiments in Figure 3c is to show the cytoplasmic localization of the nuclear localization signal deleted WDR76 mutant (WDR76 Δ NLS) and to compare its role with intact WDR76 in RAS destabilization. As we described in the manuscript, we observed that the level of RAS protein was more significantly reduced in SK-Hep1 cells overexpressing GFP-WDR76 Δ NLS compared with identical cells overexpressing GFP-WDR76 (Fig. 3c). This result agrees with the results that significant enhancement of degradation/ubiquitination of RAS by the cytoplasmic localization of WDR76 and RAS binding affinity of WDR76 Δ NLS compared with that of WDR76 full-length.

Because WDR76 degrades RAS by binding, co-localization of WDR76 and RAS has not been

expected in this experiment, and that could be observed by overexpressing catalytically inactive WDR76 mutant protein as in the experiment showing co-localization of catalytically inactive mutant Nedd4-1 C867A and RAS proteins¹.

Whereas the pattern of ubiquitination shown in Suppl. Fig. 1g is typical for poly-ubiquitination/degradation, the pattern of RAS ubiquitination observed in UbiQapture Matrix does not look like poly-ubiquitination. Could author provide an explanation for this? How does suppression of WDR76 affect the levels of ubiquitination of endogenous RAS?

Response: We agree with the reviewer's comment that the pattern of RAS ubiquitination observed in UbiQapture Matrix is different. We believe that the ubiquitination band patterns could vary depending on experimental conditions, such as reaction time, amount of reaction lysates, electrophoresis running time, and transfer conditions².

To clarify the pattern of RAS ubiquitination, we re-performed the assay using UbiQapture Matrix with an optimized condition, and confirm the consistent polyubiquitination pattern of RAS. We replaced the original data with the new data in the revised manuscript (Fig. 1h). As shown in Fig. 3f and Suppl. Fig. 4h, WDR76 loss reduced the rate of polyubiquitination-dependent proteasomal degradation of RAS. Together with the inverse correlation of WDR76 and RAS in human hepatocellular carcinoma sample, our overall data propose the role of WDR76 in the stability regulation of RAS.

Suppl. Fig. 1i: Why does RAS have different molecular weights in the lysate and after pull-down?

Response: By performing immunoprecipitation analysis, we often observed differences in the RAS band sizes as shown in Fig. 1e and 2a. The RAS band sizes are similarly different in the Supplement

Fig. 1i. However, the size difference of RAS protein appeared in the gel is more prominent due to longer time of the electrophoresis. To resolve this concern for the general readers' confusion, we performed new experiments and replaced the original gel blot with new one in the revised manuscript. .

Fig. 2d and Fig. 5f : It is also not appropriate to use a single housekeeping gene as an internal control. Normally, it should a geometric mean of several housekeeping genes.

Response: As directed by the reviewer, the geometric mean of the three internal reference genes (*β -actin*, *GAPDH*, *HPRT*) was used to correct the raw values for the genes of interest and we obtained consistent results. We replaced the original data with the new data in the revised manuscript (Fig. 2d and Fig.5f).

Fig. 4e: I am surprised that there is no variation in Ki67-staining of WDR76-positive liver tumours. I am wondering whether all tumours were exactly the same?

Response: We appreciate the reviewer for pointing out the mistake of missing error bar. We fixed the error in the graph in the revised manuscript (Fig. 4e).

Suppl. Fig3f: the key control of SK-wt cells treated with the MEK inhibitor is missed. If the tumorigenic effect of WDR76 loss is MAPK-dependent, MEK inhibition should abolish the difference between wt- and WDR-KO cells.

Response: To address this question, we performed an additional cell transformation assay using SK-WT and SK-KO cells. The MEK inhibitor AS703026 suppressed WDR76 KO-induced transformation of the SK-KO cells as well as that of the SK-WT cells, indicating that WDR76 KO-mediated RAS stabilization induces cell transformation through the MAPK pathway. We replaced the original data with the new data in the revised manuscript (Fig. 3f).

Minor points:

1. Fig.2f,g: molecular weight markers should be added

Response: We added molecular weight markers in Fig. 2f, and 2g.

2. Fig. 2d: n? what is the bar for? SD or SEM? T-test?

Response: We specified detailed information in the figure legend.

3. Fig. 2e: n? SD or SEM? Is it significant by two-way ANOVA?

Response: We specified detailed information in the figure legend.

4. Fig. 3f: molecular weight markers should be added

Response: We added molecular weight markers in Fig. 3f.

5. Fig. 4c: what does the middle line show? SD or SEM? Statistics?

Response: We specified detailed information in the figure legend.

6. Fig. 4e: number of analysed tumours; What do the bars show? SD or SEM? Statistics?

Response: We specified detailed information in the figure legend.

7. Fig.5 b,c: what does the middle line show? SD or SEM? Statistics?

Response: We specified detailed information in the figure legend.

8. Fig. 5f: number of mice? What do the bars show? SD or SEM? Statistics?

Response: We specified detailed information in the figure legend.

9. Suppl. Fig. 1g, h: molecular weight markers should be added

Response: We added molecular weight markers in Suppl. Fig. 1g, and 1h.

10. NEDD4-mediated degradation of RAS and its role in cancer development is not mentioned

<https://www.ncbi.nlm.nih.gov/pubmed/24746824>

Response: As directed by the reviewer, we have included the description of the NEDD4-mediated degradation of RAS and its role in cancer development in the Discussion section of the revised manuscript.

1. Zeng, T. et al. Impeded Nedd4-1-mediated Ras degradation underlies Ras-driven tumorigenesis. *Cell Rep.* **7**, 871-882 (2014).
2. Emmerich, C.H. & Cohen, P. Optimising methods for the preservation, capture and identification of ubiquitin chains and ubiquitylated proteins by immunoblotting. *Biochem. Biophys. Res. Commun.* **466**, 1-14 (2015).

Reviewers' Comments:

Reviewer #1:

Remarks to the Author:

I am satisfied with the revisions made.